# A qualitative interview study on quality of life and ageing experiences of autistic adults
Hannah E. Viner [1] ✉, Nicola Yuill [2], Andreia P. Costa [1], Holly Radford [3] & Anna E. Kornadt [1]

Quality of life across the lifespan has been established as a key research priority by the autism community. Still relatively little is known about the quality of life and ageing experiences of older autistic adults. Most studies to date have used generic measures of quality of life which may not accurately capture the experiences of autistic people. The aim of this qualitative study was to understand how autistic adults experience and define quality of life as they age. We conducted semi-structured interviews with 16 autistic adults aged 40+ from the UK and Luxembourg. Using reflexive thematic analysis, we identified five themes that contribute to quality of life: (1) Diagnosis is pivotal, (2) Connection with others, (3) Autonomy over space and time, (4) 'Paperwork of life', (5) Vulnerability. This study identifies autism-specific aspects of quality of life and highlights ways in which ageing and autism intersect and impact quality of life.

Autism Spectrum Condition is defined as a neurodevelopmental condition characterised by social communication differences and restrictive/repetitive behaviours and interests[1]. Although autism extends throughout life, research has focused primarily on children and young adults[2], leaving a gap in our knowledge of autistic experiences of ageing[3] (see analysis section for statement on identity-first language use). Despite an increase in research with older autistic adults, work in this area still only represents 0.4% of published autism studies over the last decade[4]. Ageing research is needed as increasing numbers of previously undiagnosed adults, the 'lost generation'[5], are being identified as autistic[6,7]. Discovering one is autistic can have a profound impact, reframing past experiences[8,9] and fostering self-understanding and acceptance[10]. Yet, autistic individuals face many unanswered questions about what ageing and late life will look like for them[11]. In this paper, we aim to better understand autistic adults' perspectives on quality of life (QoL) in the context of their ageing experiences.

To understand autistic adult development and ageing, we must first consider how autism is conceptualised and how 'successful' development for autistic adults can be defined. Historically the medical model of disability has dominated autism research, conceptualising autism as a series of deficits or impairments[12]. Adults are generally compared against a series of 'typical' levels of functioning across multiple domains with supposed shortcomings attributed to intra-individual factors or so-called 'fundamental deficits associated with autism'[13]. The autistic-led neurodiversity paradigm offers an alternative, strengths-based perspective to frame autism research[14]. It recognises autism as a 'natural and valuable form of human diversity'[15] and in line with the social model of disability[16],

highlights how existing societal structures can be disabling[17,18] and limit autistic flourishing[19].

Defining and measuring 'success' in adulthood is inextricably linked to these paradigms. Early research was rooted in the medical model, focusing on comparing autistic adults against a series of 'objective outcomes'. These were typically operationalised as a series of normative life goals such as living independently, employment, and friendships/ relationship with the majority of autistic participants deemed to have achieved 'poor outcomes'[13,20,21] (with some exceptions; see Mason et al., for review[22]).

More recently, thanks to the neurodiversity movement, autistic voices have played a greater role in this line of research[23]. In the context of premature mortality[24] and elevated rates of suicidality[25] in autistic adults, the autism community has identified the subjective measure of quality of life as a key research priority[26]. QoL is a multidimensional construct defined as an 'individual's perception of their position in life in the context of the culture and value systems in which they live and in relation to their goals, expectations, standards and concerns'[27].

Despite a shift towards this subjective measure, findings overall continue to be concerning[28]. Although there is some individual variability[29–31], numerous studies to date have reported lower levels of QoL in autistic individuals when compared to neurotypical groups[32] and individuals with other forms of neurodivergence such as ADHD[33]. Mirroring autism research in general, few studies have focused on the experiences of older adults or changes across the lifespan. In a notable exception, Van Heijst and Guerts[34] presented the first QoL study with older autistic adults (aged 50+) showing QoL was lower for adults on the spectrum compared to non-

[1]University of Luxembourg, Esch-sur-Alzette, Luxembourg. [2]University of Sussex, Brighton, UK. [3]University of Portsmouth, Portsmouth, UK.
✉e-mail: hannah.viner@uni.lu

**Table 1 | Demographic characteristics of participants**

| | UK (n = 9) n or M (SD, range) | Luxembourg (n = 7) |
|---|---|---|
| Age (in years) | 51.22 (7.89; 42-63) | 54 (6.35; 48-67) |
| Gender | | |
| Man | 3 | 6 |
| Woman | 4 | 1 |
| Non-binary | 2 | 0 |
| Nationality | | |
| British | 9 | 0 |
| Luxembourgish | 0 | 2 |
| Luxembourgish (dual nationality) | 0 | 4 |
| Dutch | 0 | 1 |
| Employment | | |
| Full-time | 3 | 4 |
| Part-time | 1 | 1 |
| Self-employed or Freelance | 5 | 0 |
| Retired | 0 | 1 |
| Unemployed | 0 | 1 |
| Household | | |
| Living alone | 2 | 2 |
| Living with friends | 1 | 0 |
| Living with spouse/partner | 4 | 0 |
| Living with spouse/partner and child(ren) | 2 | 5 |

autistic adults. In addition, the limited QoL literature with middle-aged and older autistic adults shows a negative association between QoL and mental health conditions, specifically anxiety and depression[22], and a positive association between QoL and subjective social support[35].

In line with the neurodiversity paradigm, researchers have increasingly recognised the role of external factors in autistic adults' QoL[17]. However, QoL has mostly been assessed using measures derived from the general population[36] which may not map onto autistic experiences or definitions of QoL[22,32]. It is therefore crucial to have direct input from older autistic adults, to ensure their experiences and priorities are accurately reflected in QoL measures.

McConachie and colleagues[37] approached this issue by reviewing the World Health Organization Quality of Life measure (WHOQOL-BREF[27]) and the add-on WHOQOL Disabilities module[38] in collaboration with autistic adults in the UK. This resulted in the creation of 9 additional autism-specific items (ASQoL) including topics such as autistic identity, sensory overload, friendships, and barriers to accessing services. While this measure may require further refinement[39] it provides evidence that there are indeed autism-specific aspects of QoL absent from traditional measures. McConachie and colleagues[40] further demonstrated this in a qualitative study across four countries exploring autistic views and ratings of the WHOQOL-BREF and WHOQOL Disabilities items. This led to the identification of 11 autism-specific themes relevant to autistic QoL still missing from existing measures.

Although these studies provide some insight into autistic views of QoL, the use of generic measures as the basis for discussion may have constrained their scope. Thus, autistic conceptualisations of QoL require further development. Another limitation of previous work is that most QoL studies (with a notable exception[41]) group participants from late adolescence to late life[28,42] without acknowledging potential changes in priorities or definitions of QoL across the lifespan. In gerontology literature, however, there are demonstrable shifts in priorities with age. For example, social goals change

throughout life due to perceptions of time left in life[43] or changes in external circumstances[44]. Furthermore, adaptive changes in aspirations and goals (e.g. prioritisation and potentially a reduction in a number of goals) counter the loss of biological, mental and social reserves[45]. Therefore, autistic adults' conceptualisations of QoL may also change with age.

In the context of this previous work, this study aims to understand how autistic adults experience and define quality of life as they age. We address the limitations of previous research in terms of scope, sample, and measures by using open-ended interviews with autistic adults aged 40 years and older on the topics of ageing and quality of life.

## Methods

### Sample and Recruitment
We recruited autistic adults aged 40 or over to share their views and experiences on the topics of ageing and quality of life. We opted for the relatively low minimum age of 40, considering underdiagnosis of autism in older populations and a potentially limited sample size in Luxembourg due to the country's small population. Flyers were shared in the UK (via social media, a diagnostic service, and personal contacts) and Luxembourg (via an autism organisation) inviting interested individuals to email the research team for more information. Inclusion criteria were a clinical diagnosis of autism (self-reported) and residence in the UK or Luxembourg. N = 16 adults (n = 9 UK, n = 7 Luxembourg) met the inclusion criteria, provided informed consent and took part in an open-ended interview. Participant demographic information (self-reported) is displayed in Table 1. Additionally, n = 3 participants (n = 2 UK, n = 1 Luxembourg) provided consent but dropped out as they did not meet inclusion criteria (n = 1), were too busy (n = 1), or did not respond to follow-up emails (n = 1).

The research process followed the 'Autismus-Forschungs-Kooperation' autism-friendly research guidelines[46]. All participants were sent an information sheet via email with a detailed 'what to expect' page, a short demographic questionnaire, and researcher biographies. Interview questions were provided in advance. For in-person interviews, photos and videos of the location were provided and an extra room was available for breaks.

### Measures
We used a semi-structured interview guide focused on participants' personal views and experiences of ageing and QoL. For example, we asked participants about what they were enjoying or finding challenging about ageing, what contributed or detracted from their QoL, how their QoL had changed over time and how they thought their QoL would change in the future. The full interview guide is available at https://osf.io/3cs7m/?view_only=027cb08ef0bc4e9ca6c5254a698a924a.

Interviews were conducted between August 2023 and January 2024 via video conferencing (n = 2 Luxembourg, n = 9 UK), phone (n = 2 Luxembourg) and in-person (n = 3 Luxembourg) and lasted between 40 minutes and 1 hour 48 minutes (M = 1 hour 06 minutes). The in-person interviews took place at the University of Luxembourg as per participant's choice. Using multiple forms of data collection increased the accessibility of the study. Importantly, previous work has shown that the use of multiple modalities does not impact the thematic content of interviews[47].

UK interviews were conducted in English while in Luxembourg participants chose Luxembourgish (n = 3), English (n = 3) and French (n = 1). German was also offered but not chosen. The first author was present for all interviews and conducted all English language interviews. A bilingual research assistant conducted French and Luxembourgish interviews. To ensure consistency across languages, the interviewers discussed and practised the questions and the research assistant listened to a recording of an interview carried out by the first author, prior to conducting any interviews.

### Analysis
The interviews were transcribed verbatim by the first author (English) and the research assistant (Luxembourgish and French) into NVivo 14. The data were analysed using reflexive thematic analysis, which recognises the researcher's subjectivity and active role in interpreting the data[48]. Thus, we must state our

positionality. HV is a female non-autistic doctoral student who has experience working with autistic children in both educational and research settings. HV was supervised by AK, NY and AC, female non-autistic researchers with backgrounds in gerontopsychology, psychological research in autism and learning disabilities and autism and emotion ability respectively. HR is a doctoral student who joined the team partway through the project and brings her knowledge in autism research as well as lived experience. We align with the neurodiversity paradigm and take a strengths-based approach to our research. We use predominantly identity-first language (i.e. 'autistic'), which reflects how most interviewees described themselves. This is also in line with the preferences of many in the autistic community[49]. Our analysis was informed by a critical realist understanding of language which allows us to focus on participants' lived experiences while also recognising cultural and contextual factors which influence those experiences.

During the data familiarisation stage the first author re-listened to the audio recordings and read the transcripts in their original language, identifying initial points of interest. The transcripts were coded line by line, using an inductive approach and focusing primarily on the semantic level of coding. During the next round of data review, clusters of codes with shared meaning were identified and grouped. This was an iterative process and resulted in an initial draft of themes. The draft analysis (translated into English) was reviewed by all authors. It was then adjusted based on their feedback resulting in the finalised five themes.

### Reporting summary
Further information on research design is available in the Nature Portfolio Reporting Summary linked to this article.

## Results
Reflecting on their personal experiences, contributors reported both positive and negative aspects to ageing. Positives included the accumulation of knowledge, wisdom, experience, and self-knowledge. Negatives included physical decline such as 'aches and pains' or not being able to do what they used to, as well as cognitive decline, in particular short-term memory. Participants often looked to their parents' and relatives' experiences of ageing as a reference of what to expect.

Regarding QoL, participants initially mentioned basic needs such as food, shelter, physical health, mental health, and financial stability. Beyond these items, five key themes were identified as contributing to quality of life in adulthood and during the ageing process: (1) diagnosis is pivotal: understanding and accepting self, (2) connection with others: managing your social battery, (3) autonomy over space and time: creating optimal environments, (4) 'paperwork of life': systemic barriers to quality of life, (5) vulnerability: experiences of discrimination, harassment, and bullying. An overview of the themes, sub-topics and illustrative quotes can be found in Table 2.

### Theme 1. Diagnosis is pivotal: understanding and accepting self
When asked what they were enjoying about ageing, participants repeatedly reported *'knowing myself better'* (04 UK) and being *'more comfortable with who I am'* (02 UK). All participants were diagnosed after the age of 40 (latest at age 64) and the diagnosis was described as *'transformational'* (06 UK), a moment in time when everything began to make sense. For many, the diagnosis was the culmination of years of feeling different, being seen as *'an eccentric person'* (06 UK), *'odd'* (17 UK) or *'a bit strange'* (11 LU).

*'I had been searching for years trying to understand and now voilà, it is so much better knowing myself and being able to take my needs into account.'* (10 LU).

Contributors frequently referred to personal characteristics or behaviours as *'the autistic thing of…'*, demonstrating a shift from their former labels of 'odd' to a shared sense of belonging with other autistic adults, consequently removing prior feelings of shame or failure.

*'I mean, it's reassuring often to find that other people have had the same experiences that you've had, and it does reinforce that it's not a personal inadequacy.'* (04 UK).

Furthermore, participants described being better able to identify and advocate for their needs. Self-care activities such as the need for quiet time could be perceived as *'selfish'* (01 UK) pre-diagnosis, especially if it meant taking time away from partners and children. However, recognising these as necessary for their QoL allowed people to express and engage with their needs without feeling guilty.

*'It's really about knowing myself better and taking into account my needs. I try, without feeling guilty, to put in place my routines which allow me to feel better.'* (10 LU).

Similarly, as a result of their understanding and acceptance of self, respondents reported masking (i.e. suppressing aspects of oneself to fit in[50]) less in social situations, being a more authentic version of themselves while caring less about what others think.

*'It's the whole masking thing and trying to fit in, I'm now at an age where I feel like 'sod it', I really genuinely don't have to do that anymore.'* (03 UK)

However, the benefits of diagnosis are not immediate and universal as the integration of a new autistic identity can be challenging. Participants recounted reflecting on and re-evaluating their lives, often describing a clear 'before' and 'after' diagnosis. This re-evaluation could lead to a sense of loss, wondering 'what could have been' had they been diagnosed sooner.

*'I think it's definitely impacted the first 50 years which I think is really sad because obviously there's lots of support I could have had, lots of things I wouldn't have done, wouldn't have put myself through.'* (15 UK).

This processing and accepting of the diagnosis sometimes required additional support such as therapy, coaching or extended self-reflection in which respondents confronted their initial negative responses or internalised stigma.

*'I still didn't want to be autistic because, to me it meant disabled and I didn't feel disabled.'* (17 UK).

Notably, this process did not always result in self-acceptance. Some contributors still expressed feelings of denial or disconnect from their diagnosis even many years later.

*'The syndrome, it doesn't improve anything, it deteriorates life (…) maybe it's not real, I don't fully believe in it.'* (12 LU - diagnosed 10 years prior).

### Theme 2. Connection with others: managing your social battery
Connection with others was identified as a key contributor to a good QoL. However, how this manifested varied between participants. For some, especially those with a partner and/or children, the immediate family was their primary source of connection. For others, this need could be fulfilled via the workplace, close friends, or regular interactions with acquaintances. Across participants, it was clear that socialising is draining and, importantly, that social energy is a finite resource. Therefore, having the time and space for recovery after socialising is essential for QoL.

Interviewees with a partner frequently described this person as their key (and in some cases only) connection. Partners played a multi-faceted role in their lives, often acting as a friend, support, and facilitator between the participants and *'the outside world'* (10 LU). They provide a safe space in which participants can be themselves without needing to mask. Feeling understood was a recurring sentiment.

*'My husband, very, very important and pivotal to my happiness. I think just that one stable known steady relationship where you know, we just understand each other and we're a team and that's great.'* (05 UK)

However, there were exceptions in which partners did not provide any support or had very little understanding of the autistic experience.

*'I really wish that my wife knew more, had more of an understanding about autism than she has. I think, I think it has shamefully, shamefully little meaning to her.'* (13 LU)

Those with children described them as being a source of joy as well as requiring a lot of work, often leaving little time or energy for connection with others outside the family unit.

*'They add everything, a lot of stress and anxiety and thinking but also a lot of happiness, a lot of comfort'.* (12 LU)

**Table 2 | Overview of themes and sub-topics, with illustrative quotes**

| Theme | Key sub-topics | Illustrative quotes |
|---|---|---|
| 1. Diagnosis is pivotal: understanding and accepting self. | 1.1 Understanding and accepting self | Now that I've done the late diagnosis, realising there are going to be things that I struggle with and there's a reason, I can accept it and I can actually say like I'm not, I'm not terrible. |
| | 1.2 Advocating for one's needs | Not being afraid to say no (…) I'll just say I'm autistic I don't do parties (…) it's just such a weight off my shoulders to be able to do that you know? |
| | 1.3 Masking less | I always had to play a role in order not to stand out and now I don't have to do that so much (…) I can be me. |
| | 1.4 Sense of loss / missed years | I'd be a different person, I swear (…) I may have had autistic friends or not feel bad about wearing the same clothes, liking the same food or wanting to be by yourself. |
| 2. Connection with others: managing your social battery. | 2.2 Social energy is finite | I can only cope with a little bit of socialising. I can do it and then I need a rest and I've always been like that since a child. |
| | 2.1 Partner as key | Ok so having a partner definitely was very important to me, don't have that now so that sucks, that was a big part of my stability (…) that was very much a big part of my confidence (…). |
| | 2.3 More than acquaintances, less than friends | I go and watch a lot of football matches (…) there are you know some people that I have met who also do a sort of similar thing and follow the range of local non-league teams or whatever so it's sort of a bit more than acquaintances (…). |
| | 2.4 Fear of isolation | (…) if anything happens to my husband or anything happens to me what are we going to do? (…) because [we] don't really have friends and then there'll be no family and it's kind of like oh, so I think we are worried about being completely cut off. |
| 3. Autonomy over space and time: creating optimal environments. | 3.1 Physical spaces | Having switched largely to working from home during the pandemic I knew that was brilliant for me (…) so more control like it's my room I can set it up as I like, I can have whatever lights I like, there's no noise which is amazing, I don't have to put up with other people talking or interrupting me or anything like that so that's good. |
| | 3.2 Time for interests | Just sort of broadly leisure and recreation, just having, one having the time and two having the means and then I suppose also the opportunity to be able to do things that you enjoy doing. |
| | 3.3. Interests aligned with work | I just feel really fortunate that I've found an area that I'm so passionate about, it's almost like a special interest (…) and to be able to do a job that you are so passionate about. |
| | 3.4 Predictability and routine | I think that gets harder as I get older as well in some respects so I have my routine and I love my routine but if anything goes out of my routine it can really throw me, I think there's that sense to the need for control and routine. |
| 4. 'Paperwork of life': systemic barriers to quality of life. | 4.1 Administrative tasks | I've always kind of struggled with like I describe it as like the paperwork of life (…) |
| | 4.2 Healthcare (UK) | The other fear that comes with it is if you are autistic and then you get ill and you're going to be in touch with health and services a lot more and I don't like going to the GP, I don't like being, I certainly don't like being touched. |
| 5. Vulnerability: experiences of discrimination, harassment, and bullying. | 5.1 Stereotypes and stigma | (…) it's kind of it is really hard because being autistic is so stigmatised and we've got that internalised ableism that we don't even realise is there (…) |
| | 5.2 Bullying, harassment and discrimination | (…) then I told them [the employer, about being autistic], I even showed them the diagnosis and then I was actually "promoted out of the way" meaning I got a position with a higher job title but instead of having forty-five people reporting to me I only had two people. I was going to take legal action but my lawyer said (…) as long as you don't lose any money, it's not a demotion. |

In the context of ageing, a significant fear was their partner passing away. Beyond the loss of a loved one, interviewees expressed fear of isolation.

'*My wife is really my link to the outside world and with ageing maybe my biggest worry (…) is ending up alone.*' (10 LU).

Friendships were mentioned as another source of connection. However, making and maintaining friendships was frequently described as challenging.

'*I can't make friends very well and I don't have the time and energy for them*' (02 UK).

This often led to contributors reporting few (1–3) or no friendships.

'*I always joke that when people say "autistic people don't make friends" that you only really need one, it's quite hard work*' (03 UK).

There were exceptions, with one participant stating '*I have masses of friends … I am very proud of my friendships … I have a talent for making friends*' (17 UK). The impact of having limited friendships was mixed with some participants intentionally decreasing their social interactions with age '*I'm seeking out fewer people than before*' (08 LU) while others perceived it as a gap in their life.

A third source of connection was people who are not seen as very close but provide regular interaction.

'*Acquaintances feels too stand offish but friends is probably, they don't feel close enough to me to be friends*' (05 UK).

Ideally in these interactions '*the focus isn't the social engagement, it's the incidental to some activity*' (01 UK) for example a shared interest such as football, a language class or watching their child's swimming lesson. These interactions were somewhat predictable, focusing on the shared interest and without '*enforced socialising*' (11 LU), providing a regular form of meaningful connection without the 'overhead' of maintaining a friendship.

Finally, interactions in the workplace were of value to some, although these could be draining: '*my social battery is kind of flat by the end of the day, put it that way*' (15 UK). Importantly, in the context of ageing, for those who benefit from workplace connections, retirement can result in significant social losses, '*well since I retired (…) I don't have any work acquaintances now, so that's a big social thing for me*' (11 LU).

### Theme 3. Autonomy over space and time: creating optimal environments

The environments in which participants spent their time contributed significantly to QoL. The definition of an optimal environment was personal and multidimensional but was typically related to the individual's sensory needs including having choice over food, clothing, living in a quiet location and curating personal spaces in the home. Having autonomy over these sensory aspects big or small was essential.

For those in employment, workplaces could be particularly challenging, especially if participants did not disclose that they were autistic. This was particularly noticeable in Luxembourg (although there were exceptions) leading to contributors spending significant amounts of time in overwhelming environments.

*'It is open space where we are all together, there is a lot of noise. [If I disclosed] I could perhaps ask to be able to work alone, perhaps not all the time but at least an office in which I can be alone for part of the day for example.'* (10 LU).

When discussing how they spent their time, participants regularly evoked the notion of autonomy, in both their personal and, when applicable, work life. Echoing the statements from themes 1 and 2, time alone was a fundamental contributor to QoL. This time was frequently spent on interests (e.g., urban exploring, reading, recording butterflies). Rather than simple hobbies these were lifelong passions which were both a source of joy but also rest and recovery and were usually (although not always) enjoyed alone. Therefore, participants emphasised the importance of having the time and means to engage in these activities for their QoL.

In terms of time management, routine, predictability and planning were repeatedly discussed as beneficial for QoL. Unpredictability in any context whether it be daily routines, social interactions or going to new places was described as challenging. Planning and research were presented as a strategy allowing participants to step out of their comfort zones and cope with new situations.

*'I like everything to be the same and I don't like it when things change. I can cope with change, but I like to prepare for it so like going to [a European country, removed for pseudonymisation] last year, I decided it would be a good thing to hire a car (…) so I watched a whole video, I'd never driven on the wrong side of the road.'* (17 UK).

Finally, for those in employment, freedom and autonomy in work tasks were associated with better QoL. This was particularly true for those whose work aligned with their interests with participants describing purpose and high levels of satisfaction in their work. When this was not the case, time spent at work could be incredibly detrimental to interviewees' QoL with one participant who was *'going to work every day to do something I don't want to do'* (07 LU) recounting how this was all-consuming, negatively affecting every aspect of his life *'so severely that I can't concentrate on anything'* (07 LU).

With ageing, contributors expressed profound concerns about moving into a care home, framing it as the antithesis to autonomy and control over one's space and time. Interviewees' opinions were often informed by their experiences visiting relatives in care homes. The sensory aspects, forced socialisation and shared accommodation were perceived as being incredibly detrimental to QoL.

*'The noise and the interactivity and the loss of sort of autonomy and the forced social engagement (…) I would just, I see zero quality of life in that, in that sort of situation'* (01 UK).

### Theme 4. 'Paperwork of life': systemic barriers to quality of life

*'I'm very much allergic to any form of administrative work, it makes my blood boil in an instant.'* (13 LU).

The *'paperwork of life'* (06 UK) was reported to significantly detract from QoL. These inevitable administrative tasks such as *'filling in forms'* (02 UK), *'car insurance'* (06 UK), or *'engagement with contractors'* for house repairs (01 UK), were frequently experienced as inaccessible and challenging. The difficulties involved in completing these tasks can lead to avoidance and consequently negatively impact QoL, such as a home *'becoming shabbier and more shambolic'* (01 UK) or perhaps more seriously, delaying

medical care due to administrative barriers. Participants identified family members or partners as a source of support for these tasks. However, there were significant concerns about how they would cope in the future if their relative were to pass away.

*'It's taken a lot for my husband to try and encourage me to develop that because he's worried about what will happen if he's not there because I really struggle in this area, I struggle to do fill in forms and stuff …'* (02 UK).

The topic of barriers to healthcare was repeatedly raised in the UK sample as a significant challenge for QoL. Interviewees reported struggling to identify when they were unwell and in need of healthcare. This could lead to life-threatening conditions going undetected.

*'Before I got diagnosed with autism, I got sepsis (….). It was caught right at the last minute, I was very lucky, erm but that all came from the fact that I didn't even know I was ill.'* (06 UK).

In addition, healthcare settings frequently involve overwhelming sensory experiences which could lead to avoidance of seeking care. Finally, previous negative experiences or trauma were also sources of avoidance and fear. One participant recounted her experience in a mental health setting in the UK.

*'It terrified me, and it terrifies me now that if I were ever to have a mental health crisis or a mental health issue, the first thing they would do is lock me up in a place like that.'* (02 UK).

Overall, these barriers can lead to adults living with chronic conditions which are inevitably detrimental to QoL. Regarding ageing, contributors recognised the likelihood of medical issues increasing and therefore more frequent interactions with healthcare services.

### Theme 5. Vulnerability: experiences of discrimination, harassment, and bullying

Participants were regularly confronted with stereotypes and stigma related to autism in a multitude of contexts, reporting feelings of being misunderstood or patronised and illustrating the lack of knowledge and understanding of the general public. These involved receiving questions such as *'how can you be autistic and married?'*, *'how can you be autistic with children?'* (03 UK) or being exposed to stereotypes of autistic people as savants with references to *'Rain Man'* (09, 10 LU). In fact, misconceptions about autism were experienced as so prevalent that some participants had internalised them, leading to them dismissing the idea that they could be autistic, further delaying diagnosis.

*'It's the cliché, really, a person like Rain Man in the movies. (…) That's the cliché I had in my mind, so I never looked into autism as such.'* (10 LU).

Another implication of this stigma was a reluctance to disclose both in personal and professional contexts, with some describing negative experiences resulting from disclosure.

*'There's been actual direct discrimination, I've had people talk to me slowly as if I'm stupid. These are people that have worked with me for two years and suddenly they're directly repeating themselves as if I can't understand them.'* (05 UK).

Beyond stigma, many participants shared experiences of bullying and harassment, both verbal and sexual. Participants described how autistic experiences can contribute to increased vulnerability.

*'I do get exposed to a fair degree of harassment just through sort of being on my own, especially from, you know, as you might expect, kids aged, you know, 10 to 15 who will sort of pick on the, the older vulnerable person wandering around.'* (01 UK).

*'I think that being autistic has something to do with that because (…) I was never part of gossip, I was never part of sort of cliques, so I think that other people might have been told on the quiet that certain people were risky and dangerous.'* (04 UK).

Overall *'a lot still needs to be done in society so that autistic people are not discriminated against.'* (09 LU).

## Discussion

We aimed to understand how autistic adults define and experience quality of life as they age. Factors contributing to QoL included enhanced

self-understanding through diagnosis, the important, yet nuanced, role of connection with others, the need to create optimal environments and autonomy over how time is spent, systemic barriers to QoL, and experiences of discrimination, harassment, and bullying.

The multifaceted emotional impact of diagnosis expressed by our interviewees is consistent with previous research on diagnosis in adulthood[51–53]. First-person accounts of living as an undiagnosed autistic adult demonstrate how a lack of diagnosis can negatively impact one's QoL including negative self-conceptions[9], mental health issues[8–10], and trauma[9]. A diagnosis however, can profoundly change one's sense of self[9], offering explanations for past experiences[8] and promoting self-compassion[10]. The benefits of being recognised as autistic highlight the value of diagnosis at any stage of the life course and the potential positive impact on QoL.

Age of diagnosis is a critical factor in QoL, with later diagnosis correlating with poorer QoL[54]. In England alone, it is estimated that between 250,000 and 600,000 people aged 50+ may be autistic and undiagnosed[55]. However those seeking a diagnosis face multiple barriers, including previous misdiagnosis, extensive wait times and financial costs (see Huang et al., for review[53]). Increased resources are, therefore, urgently needed to improve adult diagnostic services. Notably, in our sample, diagnosis alone did not guarantee self-acceptance. This is significant given greater personal autism acceptance predicts lower depressive symptoms[56], and positive autistic identity or autism pride is associated with increased self-esteem[57,58]. Post-diagnostic support can facilitate the integration of one's autistic identity, focusing on autistic strengths rather than autism as a deficit[12]. At present, post-diagnostic support has been found to be lacking in the UK[59] and across Europe[60].

Post-diagnosis and with age, participants reported masking less. Masking or 'camouflaging' is a common coping strategy used to 'fit' into a neurotypical world. However, the negative impacts on mental health are well known through qualitative[61] and quantitative work[62,63]. More studies are needed to establish if this reported reduction in masking is significant and how it could impact mental health in later life.

Connection with others was identified as another key theme contributing to QoL. Similarly to studies with younger autistic people, participants expressed a desire for connection with others, yet many reported difficulties in forming and maintaining relationships[64]. Our findings add to the limited literature on social connection in older autistic adults, showing multiple and varied sources of connection, beyond family and friends. Experiences of loneliness across the lifespan have been documented both qualitatively[64,65] and quantitatively[66] with feelings of loneliness increasing with age. Even so, older autistic adults rate their social QoL as significantly better than younger autistic adults do[41]. Therefore, the relationship between social QoL and day-to-day experiences of loneliness may differ at different life stages and thus warrants further work. Studies on social relationships in neurotypical populations identify two distinct networks; the 'global network' which includes all social relationships and the 'personal network' comprised of closer personal relationships (e.g. family and friends)[67]. When assessing the social domain of QoL, the WHOQOL-BREF[27] and ASQoL[37] focus on 'personal networks' specifically 'personal relationships', 'friendships' and 'support'. However, our results and those of Chan and colleagues[68] suggest a more nuanced picture of social connection, highlighting the need to reassess how we define and measure social QoL.

Participants raised concerns about the future loss of their partner or spouse, demonstrating the need for further work on the impact of these life events on autistic adults' QoL. Considering the challenges many autistic people experience creating and maintaining social relationships, further support may be required to avoid isolation. Crucially, autistic voices should be central to designing and developing such interventions[69]. It is also important to recognise the role of non-autistic people in cross-neurotype interactions (see 'the double empathy problem'[70]) with negative perceptions of autistic people and a lack of acceptance contributing to breakdowns in mutual understanding[71].

Having autonomy and control over physical environments was identified as a key contributor to QoL, consistent with findings in autistic young adults[72]. This was particularly salient in the workplace, especially if participants had not disclosed they were autistic and consequently did not have adjustments in place. Quantitative work has identified employment as a positive predictor of QoL[73]; however, poor person-environment fit, lack of support, social stigma and negative disclosure experiences contribute to difficulties in gaining and maintaining employment[74]. While disclosure in the workplace can lead to acceptance and adjustments, those who disclose face the risk of discrimination and stigma[75]. Importantly, adults diagnosed later experience higher perceived discrimination upon disclosure than those diagnosed at a younger age[76]. These findings highlight a gap in current employment provisions. At the individual level support is needed on how to advocate for one's needs, rights to adjustments and how and when to disclose in the workplace, especially for late-diagnosed adults. However, the onus should not be on autistic individuals to ensure positive work experiences. At the employer level there is a clear need for further education, acceptance and support policies to create inclusive environments in which autistic adults can thrive.

The 'paperwork of life' was an additional negative contributor to QoL which is not fully captured in previous QoL measures. Administrative tasks were especially challenging, with adults frequently relying on informal support (e.g., parents, spouse) to access important services. Complex bureaucratic systems serve as a significant barrier to accessing state support and resources (e.g. benefits)[77]. Navigating such procedures demands significant cognitive resources such as executive functioning[78], which can be particularly challenging for autistic adults[79]. As a result, those most in need of support are disproportionately affected, further exacerbating existing inequalities[78].

In addition, contributors from the UK reflected on inaccessible healthcare services, a key example of where autism and ageing converge. Ageism and age stereotypes are ubiquitous but are of particular concern in healthcare settings as they can shape practitioners' views on treatment[80] and lead to patronising and ineffective interactions[81]. Autistic adults frequently experience barriers to healthcare[82–84] and while solutions to these challenges have been proposed[85–87], two key areas warrant further work. Firstly, it is of note that in our sample, challenges accessing healthcare were exclusively mentioned by UK participants. Considering the variability in healthcare systems globally, cross-cultural studies could be of value to identify challenges and best practices. Secondly, research is also required to assess other late-life service provisions such as residential care. While initial steps have been taken to engage the autism community in priority-setting on this topic[88], little is known about the lived experience of older autistic adults in these settings.

The final factor for QoL identified in our study was experiences of discrimination, bullying and harassment. Consistent with previous work, participants described multiple incidences of victimisation[89,90] across the lifespan. Recent work[91] has proposed applying the Minority Stress Model[92] to recognise the additive impact of these events on mental health difficulties, which are elevated in autistic populations[93]. The model recognises autistic people as a minoritised group who are repeatedly exposed to stigma-related stressors, such as the harassment and bullying reported by our participants. Such events may be exacerbated with increasing age, as individuals might additionally become targets of ageism, for instance in the healthcare sector[80,81,94]. Future work should utilise intersectional frameworks to better understand the impact of belonging to multiple marginalised groups. Finally, more interventions are required to ensure autism awareness translates into autism acceptance in non-autistic populations.

These results incrementally add to our understanding of QoL of autistic individuals across the life span. While previous work suggests that some themes, such as social connection, are also important for young adults, these topics may gain differential importance with the changing contexts of adulthood.

Reviewing our findings through the lens of the neurodiversity paradigm[12,14,15], it is evident that autistic QoL goes far beyond intra-individual factors and is shaped by the broader contexts in which autistic people live. This study demonstrates that autistic adults continue to face a

multitude of obstacles in their daily lives including inaccessible processes (e.g. administrative tasks), inaccessible environments (e.g. work places, healthcare settings, residential care) and are confronted with stigma, stereotypes and discrimination. When assessing QoL it is essential to move away from the neurotypical ideal as a goal but rather consider autistic conceptualisations of QoL. Furthermore, we must recognise that difficulties obtaining a 'good' QoL cannot be alleviated by simply addressing individual factors but requires broader systemic change.

## Limitations

This study makes a valuable contribution to the existing literature casting light on autistic experiences of ageing and QoL. We address limitations of previous research in terms of scope, sample, and measures by using open-ended interviews with autistic adults aged 40 and over. Nevertheless, there are several aspects that could be improved in future work. Firstly, differing recruitment strategies in the UK and Luxembourg may have resulted in sampling bias. Although we did not collect data on professional roles, it was noted that participants who were recruited via Twitter/X were more likely to be involved in academia and/or autism activism and advocacy. We did not collect data on ethnicity or socio-economic status which limits knowledge of our sample. We did not include self-diagnosed adults who may have represented a wider range of experiences. Given the nature of a qualitative study, sample size is relatively small and did not allow further differentiation in terms of age, gender or occupational status of our participants. Furthermore, the lower age limit for our inclusion criteria was 40 years rather than the recommended age of 50[95], and our oldest participant was only aged 67. As such, there may be differences between the views of middle-aged and older-aged adults which were not captured in this study. Regarding data collection, the interviews were conducted by two different researchers. Thus, it is possible there were subtle differences in the way questions were asked or differences in participant-researcher rapport. Regarding analysis, researcher background and theoretical viewpoints (disclosed in the positionality statement), will have influenced our interpretations of the data. Further involvement of autistic researchers in question construction and analysis would also be beneficial. Finally, the present sample did not include any autistic adults with a learning disability, for which other methods would be needed.

## Conclusions

This study provides evidence of autism-specific factors of QoL and highlights ways in which ageing and autism can intersect, for better or for worse, in later life. Further work is required to create QoL measures which are relevant and meaningful to autistic adults. It would also be beneficial to conduct longitudinal lifespan research to better understand how QoL and its determinants change (or remain the same) as people get older. Finally, future research should focus on further identifying structural and societal barriers to autistic QoL and on co-designing meaningful solutions that support adults across the lifespan to obtain and maintain high levels of QoL, however they choose to define it.

## Data availability

The full interview guide is available at https://osf.io/3cs7m/?view_only=027cb08ef0bc4e9ca6c5254a698a924a. Neither the raw nor aggregate data used in this study are publicly available as they contain information that could compromise the privacy of research participants (ERP 23-035 QoLAA).

## Code availability

No code was used to produce the result.

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

## Acknowledgements

The authors would like to acknowledge the contributions of Cathy Hoffmann who assisted with interviewing and transcription, the Fondation Autisme Luxembourg for their support with recruitment and, most of all, our study participants. Funded by the Luxembourg National Research Fund (Ref Nr. 17027176). The funders had no role in study design, data collection and analysis, the decision to publish or preparation of the manuscript.

## Author contributions

H.E.V.: Conceptualisation, funding acquisition, methodology, investigation, formal analysis, writing (original draft preparation). N.Y.: Methodology, review of analysis, writing (review and editing) and supervision. A.P.C.: Review of analysis, writing (review and editing) and supervision. H.R.: Review of analysis and writing (review and editing). A.E.K.: Conceptualisation, methodology, review of analysis, writing (review and editing) and supervision (lead).

## Competing interests

The authors declare no competing interests.

## Ethics

Ethical approval for this project was granted by the University of Luxembourg Ethics Committee (ERP 23-035 QoLAA). This study was not preregistered.
