## [Peer review File · Communications Psychology]

A qualitative interview study on quality of life and ageing experiences of autistic adults

Corresponding Author: Ms Hannah Viner

Version 0:

Decision Letter:

Dear Mx Viner,

Thank you for your patience during the peer-review process. Your manuscript titled "Quality of Life and Ageing Experiences of Autistic Adults Aged 40+: a Qualitative Interview Study." has now been seen by 3 reviewers, and I include their comments at the end of this message. They find your work of interest but raised some important points. We are interested in the possibility of publishing your study in Communications Psychology, but would like to consider your responses to these concerns and assess a revised manuscript before we make a final decision on publication.

We therefore invite you to revise and resubmit your manuscript, along with a point-by-point response to the reviewers. Please highlight all changes in the manuscript text file.

Editorially, we consider it important that you situate your findings within relevant theoretical frameworks as suggested by Reviewer 3. We also encourage you to expand on methods and limitations as requested by Reviewer 1 and 3 respectively.

I am attaching an Editorial Requests Table that details critical reporting requirements for the revised manuscript. Please attend to each item and ensure your manuscript is fully compliant. We are requesting that your manuscript aligns with these requirements as this facilitates the evaluation of your manuscript, reducing delays in re-review and potential future acceptance. If your revised manuscript is not aligned with these requests on major issues, such as those concerning statistics, it may be returned to you for further revisions without re-review. Additional information can be found in our style and formatting guide Communications Psychology formatting guide.

Please use the following link to submit your

- revised manuscript,
- point-by-point response to the referees' comments,
- cover letter (as a separate document),
- the Editorial Policy Checklist (see below),
- the Reporting Summary (see below), and
- the completed Editorial Request Table (attached):

Link Redacted

Best regards,

Jennifer Bellingtier

Jennifer Bellingtier, PhD
Senior Editor
Communications Psychology

REVIEWER EXPERTISE:

Reviewer #1 autism, qualitative research, lifespan development

Reviewer #2 autism, lifespan development

Reviewer #3 autism, qualitative research

REVIEWER REPORTS:

Reviewer #1 (Remarks to the Author):

I wish to thank the authors for a thoughtful, well designed, and well written piece that adds to the sparse literature on quality of life in older autistic adults - a very valuable research area.

I have queries, more so than comments, on this manuscript that may result in minor edits.

1. The authors state "We opted for the relatively low minimum age of 40, taking into account the 88 underdiagnosis of autism in older populations and a potentially limited sample size in Luxembourg.", purely for interest of the reader, could the authors very briefly extend this sentence to indicate the reason(s) for expecting a limited sample size in Luxembourg (that directly relates to the decision to set the minimum age for the sample at 40)?

2. The authors note that a limitation of the study is that it does not capture a 'full range' of experiences of older autistic adults due to sampling issues in this age range. Did the authors ever consider including self-diagnosed individuals, given this issue (and if so, could they possibly add a line or so on this in the manuscript)?

3. In taking a look at the interview schedule, I noted the question 'In general, what do you think quality of life means'. I'm glad to have seen this included, since it's key to ascertain whether participants and researchers are communicating about the same concept. I would be interested to understand in a broad sense whether the participant responses to this question included factors that align with the existing definitions of QoL (I get the sense they did in the opening of the Results section)? In addition, I noticed the question about rating QoL from 1-10 and was interested in what participants responded here also.

Reviewer #2 (Remarks to the Author):

Dear authors,

Thank you for reviewing your interesting article about autistic adults. First, it is a well-written paper. The rationale for the study is well explained. Also, the method-section is thoroughly explained in line with Braun & Clarke (2022).

The themes seem to fit the description and illustrated with useful quotes. You could consider creating a table that shows examples of how you developed the themes. You can find an example of how this is done in source 55 in your literature list. It is also possible to add a figure that illustrates the themes.

I see that you have structured the paper with a section named "analysis" where you give an account for the themes and following quotes. Subsequently, you have the discussion section. It is also possible to merge these two sections, so that the discussion is written below each theme in the analysis part. This is for example done in this article:

<https://doi.org/10.1016/j.ssmqr.2022.100100>.

The limitation section was more in line with a quantitative study. You can write more about the limitations that has to do with a qualitative interview study, such as different persons conducting the interviews, bias in the interview guide, or how your background shaped the way you analysed the findings.

Overall, the manuscript is well-written and highlight some interesting findings.

Reviewer #3 (Remarks to the Author):

This manuscript reports on a qualitative study with late-diagnosed autistic adults (aged 40+ years) about their quality of life. Understanding what it means for autistic people to live well has not been properly investigated by autism researchers, largely owing to the dominance of a deficits-based, medical model. The focus on discovering what matters to autistic people – especially older autistic people – is therefore timely and important.

I enjoyed reading this manuscript, especially because I felt the authors had presented a sophisticated analysis of participants' responses.

Nevertheless, I was disappointed primarily because the findings were not situated within a broader theoretical context. Normative frameworks for quality of life were not discussed. Nor were the many broader frameworks within positive psychology, political philosophy, economics, sociology and related disciplines, which provide a thorough consideration of what it means to have a good human life, and what it might take for people to achieve good quality of life, on their own terms. This is especially important in the case of autistic people, when their autonomy in this sense is so often denied. Some of these frameworks have recently been applied to autistic people (see Chapman & Carel, 2022, *Journal of Social Philosophy*; Pellicano et al., 2022, *Nature Reviews Psychology*), which highlight some of the very issues discussed in the current review (especially regarding social connection and autonomy). This manuscript would be strengthened by thoroughly considering such conceptual issues.

Considering broader conceptual notions of a good autistic life is also important because such frameworks emphasise that people's life chances (i.e., the opportunities each individual has to improve their quality of life) are shaped by a range of factors *beyond the person*, including within families, schools, workplaces, communities and society. This focus on supports that go beyond the individual person was emphasised in Themes 3, 4 and 5. Incorporate such thinking in the current manuscript should therefore contribute to discussions about how older autistic people can be supported – by adapting the environment, rather than the person.

My second concern was that Theme 1 related strongly to participants' experiences of late diagnosis, but the authors did not review much of the literature on late diagnosis. Key references (e.g., Leedham et al., 2020; Lilley et al., 2022; Stagg & Belcher, 2019) were missing, which made this part of the manuscript the weakest. It could be strengthened by more foreshadowing of the experiences of late-diagnosed autistic people in the introduction, and greater discussion of how such experiences contributed to perceptions of quality of life in the discussion.

In short, I felt that this study could eventually make a significant contribution to the literature, but much work needs to be done to integrate the research and the findings into the relevant theoretical and empirical literature.

Minor comments:

1. Did the authors record when (what age) people received a formal diagnosis of autism or first self-identified? It would be helpful to include this information in the manuscript.
2. Were all the transcripts translated into English? Did any of the researchers read all transcripts? This could be clearer.

EDITORIAL POLICIES

We ask that you ensure your manuscript complies with our editorial policies and reporting requirements.

To that end, we require revised manuscripts to be accompanied by two completed items: a reporting summary that collects information on study design and procedure, and an editorial policy checklist that verifies compliance with all required editorial policies.

- <https://www.nature.com/documents/nr-reporting-summary.zip>>Nature Research Reporting Summary
- <https://www.nature.com/documents/nr-editorial-policy-checklist.pdf>>Editorial Policy Checklist

All points on the policy checklist must be addressed. Your revised manuscript can only be sent back to the referees if these checklists are completed and uploaded with the revision.

Notes: If you have submitted a Stage 1 Registered Report, Review, Primer, Comment, or Perspective you do not need to submit these forms. If you have already submitted these forms, you may disregard this request.

* **TRANSPARENT PEER REVIEW:** Communications Psychology uses a transparent peer review system. This means that we publish the editorial decision letters including Reviewers' comments to the authors and the author rebuttal letters online as a supplementary peer review file. However, on author request, confidential information and data can be removed from the

published reviewer reports and rebuttal letters prior to publication. If your manuscript has been previously reviewed at another journal, those Reviewers' comments would not form part of the published peer review file.

Communications Psychology is committed to improving transparency in authorship. As part of our efforts in this direction, we are now requesting that all authors identified as 'corresponding author' create and link their Open Researcher and Contributor Identifier (ORCID) with their account on the Manuscript Tracking System prior to acceptance. ORCID helps the scientific community achieve unambiguous attribution of all scholarly contributions. You can create and link your ORCID from the home page of the Manuscript Tracking System by clicking on 'Modify my Springer Nature account' and following the instructions in the link below. Please also inform all co-authors that they can add their ORCID to their accounts and that they must do so prior to acceptance.

Author Rebuttal letter: The author's response to these comments can be found at the end of this file.

Version 1:

Decision Letter:

Dear Mx Viner,

Your manuscript titled "Quality of Life and Ageing Experiences of Autistic Adults Aged 40+: a Qualitative Interview Study." has now been seen by our reviewers, whose comments appear below. In light of their advice I am delighted to say that we are happy, in principle, to publish a suitably revised version in Communications Psychology.

We therefore invite you to revise your paper one last time to address the remaining concerns of our reviewers and a list of editorial requests. At the same time we ask that you edit your manuscript to comply with our format requirements and to maximise the accessibility and therefore the impact of your work.

EDITORIAL REQUESTS:

SUBMISSION INFORMATION:

OPEN ACCESS:

* TRANSPARENT PEER REVIEW: Communications Psychology uses a transparent peer review system. On author request, confidential information and data can be removed from the published reviewer reports and rebuttal letters prior to

publication. If you are concerned about the release of confidential data, please let us know specifically what information you would like to have removed. Please note that we cannot incorporate redactions for any other reasons.

Link Redacted

Best regards,

Marika Schiffer

Marika Schiffer, PhD
Chief Editor
Communications Psychology

REVIEWERS' COMMENTS:

Reviewer #1 (Remarks to the Author):

I wish to thank the authors for their considered responses to the original review process. I am happy with the changes made, and wish the authors all the best with their manuscript.

Reviewer #2 (Remarks to the Author):

Dear authors,

I have read through your responses, and I believe the manuscript has been improved accordingly.

All the best.

Reviewer #3 (Remarks to the Author):

The authors have done an excellent job of responding to the reviewers' comments and revising their manuscript accordingly.

I have three additional comments for the authors to consider.

First, while the aim of this paper was to understand how older autistic adults experience and define quality of life, the discussion could benefit from the authors being more explicit about whether the current findings corroborate the findings from existing studies, mostly on younger autistic adults. That is to say, are the five themes identified – Diagnosis is pivotal, Connection with others, Autonomy over space and time, 'Paperwork of life', and Vulnerability – characteristic of autistic quality of life across the lifespan or are there any specific findings /aspects of quality of life that appear to be particularly important for (or unique to) older autistic adults? The authors do make these comparisons at some points (e.g., discrimination may be exacerbated with increasing age, as individuals encounter ageism, p. 13), but more explicit discussion of this issue would be helpful.

Second, on the theme titled, 'paperwork of life', much has been written on the overly bureaucratic nature of benefits systems and the stigma it can cause disabled people (e.g. see Saffer et al., 2018, Disability & Society). There is also some work on how this (relentless) form-filling can be particularly disadvantageous for disabled people – and therefore autistic people – because too many resources (executive functions) are required to navigate the systems necessary to secure such assistance (see Christensen et al., 2020, Public Administrative Review). The discussion of this theme (see p. 13) would be strengthened by reference to this work.

Finally, for the new paragraph on pp. 13 and 14 beginning "Reviewing our findings through the lens of the neurodiversity paradigm...", it would be helpful to include some references, not least to demonstrate how the current findings build upon existing theoretical and empirical work.

Author Rebuttal letter: The author's response to these comments can be found at the end of this file.

Reviewer 1

I wish to thank the authors for a thoughtful, well designed, and well written piece that adds to the sparse literature on quality of life in older autistic adults - a very valuable research area.

We thank the reviewer for this positive evaluation of our study.

R1_1. *The authors state "We opted for the relatively low minimum age of 40, taking into account the underdiagnosis of autism in older populations and a potentially limited sample size in Luxembourg.", purely for interest of the reader, could the authors very briefly extend this sentence to indicate the reason(s) for expecting a limited sample size in Luxembourg (that directly relates to the decision to set the minimum age for the sample at 40)?*

The main concern in Luxembourg was the country's small population, which currently sits at 675,050. This, combined with autism prevalence rates at approximately 1% and underdiagnosis in older adults, suggested a potentially limited sample size. We have extended the sentence in the manuscript to clarify this reasoning (p. 3).

R1_2. *The authors note that a limitation of the study is that it does not capture a 'full range' of experiences of older autistic adults due to sampling issues in this age range. Did the authors ever consider including self-diagnosed individuals, given this issue (and if so, could they possibly add a line or so on this in the manuscript)?*

During the planning phase of this study, we did not consider including self-diagnosed individuals. Having learnt more since then about the barriers to getting a diagnosis and how these may disproportionately affect underrepresented groups, we will include self-diagnosed individuals in future work. We have added this to the limitation section (p. 14).

R1_3. *In taking a look at the interview schedule, I noted the question 'In general, what do you think quality of life means'. I'm glad to have seen this included, since it's key to ascertain whether participants and researchers are communicating about the same concept. I would be interested to understand in a broad sense whether the participant responses to this question included factors that align with the existing definitions of QoL (I get the sense they did in the opening of the Results section)?*

In light of your comment, we went back to the transcripts and reviewed participants' responses to this question. Their views are indeed reflected in the opening of the results section, as noted. Participants listed basic needs (food, shelter), physical health, mental health, financial stability. Additional items that came up were time for leisure, autonomy, and family and friendships. These latter items were discussed much more extensively and are captured in the themes.

R1_4. *In addition, I noticed the question about rating QoL from 1-10 and was interested in what participants responded here also.*

The rating question elicited a variety of answers. Seven participants gave an exact number (ranging from 2/10 to 9/10). Four participants gave no answer for reasons such as "I find it very difficult in terms of putting a number on it because it would change" or "well that's a nonsense question". Finally, five participants gave multiple answers because "it depends" or was context specific (e.g. 4/10 at work but 8/10 at home). It was therefore challenging to include this data in a meaningful way, and as our study mainly focused on how autistic adults define quality of life, we decided not to report this information.

Reviewer 2

Thank you for reviewing your interesting article about autistic adults. First, it is a well-written paper. The rationale for the study is well explained. Also, the method-section is thoroughly explained in line with Braun & Clarke (2022).

We thank the reviewer for this positive evaluation of our work.

R2_1. *The themes seem to fit the description and illustrated with useful quotes. You could consider creating a table that shows examples of how you developed the themes. You can find an example of how this is done in source 55 in your literature list. It is also possible to add a figure that illustrates the themes.*

Thank you for your suggestion and for referencing an example table in source 55. We have added a table as an Appendix to show examples of how the themes were developed. The table also include additional quotes. As we do not have specific subthemes, we added the identified sub-topics to the table, as they are discussed in the manuscript.

R2_2. *I see that you have structured the paper with a section named "analysis" where you give an account for the themes and following quotes. Subsequently, you have the discussion section. It is also possible to merge these two sections, so that the discussion is written below each theme in the analysis part. This is for example done in this article: <https://doi.org/10.1016/j.ssmqr.2022.100100>.*

Thank you for the suggestion. We did consider adopting a more integrated structure in which the analysis and results are written together, however we ultimately opted to separate them. We thought it provided more clarity, particularly for researchers who work with more quantitative methods and who may be interested in our work.

R2_3. *The limitation section was more in line with a quantitative study. You can write more about the limitations that has to do with a qualitative interview study, such as different persons conducting the interviews, bias in the interview guide, or how your background shaped the way you analysed the findings.*

Thank you for highlighting this. We have now acknowledged further limitations which are more pertinent to qualitative research. These include acknowledging the potential impact of having two different researchers conduct the interviews and the influence of our backgrounds and theoretical viewpoints on the interpretation of the data (p. 14).

Reviewer 3

This manuscript reports on a qualitative study with late-diagnosed autistic adults (aged 40+ years) about their quality of life. Understanding what it means for autistic people to live well has not been properly investigated by autism researchers, largely owing to the dominance of a deficits-based, medical model. The focus on discovering what matters to autistic people – especially older autistic people – is therefore timely and important. I enjoyed reading this manuscript, especially because I felt the authors had presented a sophisticated analysis of participants' responses.

Thank you for this positive evaluation of our work.

R3_1. *Nevertheless, I was disappointed primarily because the findings were not situated within a broader theoretical context. Normative frameworks for quality of life were not discussed. Nor were the many broader frameworks within positive psychology, political philosophy, economics, sociology and related disciplines, which provide a thorough consideration of what it means to have a good human life, and what it might take for people to achieve good quality of life, on their own terms. This is especially important in the case of autistic people, when their autonomy in this sense is so often denied. Some of these frameworks have recently been applied to autistic people (see Chapman & Carel, 2022, *Journal of Social Philosophy*; Pellicano et al., 2022, *Nature Reviews Psychology*), which highlight some of the very issues discussed in the current review (especially regarding social connection and autonomy). This manuscript would be strengthened by thoroughly considering such conceptual issues.*

*Considering broader conceptual notions of a good autistic life is also important because such frameworks emphasise that people's life chances (i.e., the opportunities each individual has to improve their quality of life) are shaped by a range of factors *beyond the person*, including within families, schools, workplaces, communities and*

society. This focus on supports that go beyond the individual person was emphasised in Themes 3, 4 and 5. Incorporate such thinking in the current manuscript should therefore contribute to discussions about how older autistic people can be supported – by adapting the environment, rather than the person.

Thank you for this feedback and for the detailed examples and references you provided. We have made multiple changes based on your comments.

We have made significant changes to the introduction to better position our study within existing theoretical frameworks. More specifically, we have introduced the medical model and neurodiversity paradigm of autism and have explored how they influence our perceptions of what 'successful adulthood' looks like. We have highlighted how early studies focused on 'objective outcomes', operationalised using normative standards of 'success' (pp. 1-2). We further describe the problems associated with this approach, such as the deficit-based views of autism which attributed so-called shortcomings to individual factors (p. 2). We have also discussed quality of life as a subjective assessment of adulthood and have highlighted the lack of autistic voices in the conceptualisation of QoL.

We have also made changes to the discussion section. To further acknowledge and emphasise how external factors influence life chances we have added information about barriers to diagnosis, a reference to the double-empathy problem and emphasised the role of the employer in workplace difficulties (pp. 12-13). We have also added a final paragraph in the discussion to summarise our findings within the neurodiversity paradigm (p. 14).

We think the manuscript has benefitted immensely by these additions, however, we also note, that the scope of the manuscript and the availability of space does not allow us to delve into the topics mentioned in more depth.

R3_2. *My second concern was that Theme 1 related strongly to participants' experiences of late diagnosis, but the authors did not review much of the literature on late diagnosis. Key references (e.g., Leedham et al., 2020; Lilley et al., 2022; Stagg & Belcher, 2019) were missing, which made this part of the manuscript the weakest. It could be strengthened by more foreshadowing of the experiences of late-diagnosed autistic people in the introduction, and greater discussion of how such experiences contributed to perceptions of quality of life in the discussion.*

In response to your feedback, we have added a sentence in the introduction about the impact of diagnosis (p. 1). We have added a particularly relevant quote in the results section which aligns with the recommended papers (p. 6). We have expanded the discussion of theme 1, including the literature you provided and have reflected on how diagnosis may impact QoL (p. 12). In addition to further emphasise systemic barriers to QoL, we have highlighted the multiple challenges faced by those seeking a diagnosis (p. 12).

R3_3. *Did the authors record when (what age) people received a formal diagnosis of autism or first self-identified? It would be helpful to include this information in the manuscript.*

Unfortunately, we did not record the specific ages at which participants received their formal autism diagnosis. During the interviews participants mentioned when they were diagnosed, which allowed us to determine that all participants were diagnosed between the ages of 40 and 64. However, the data provided, such as participants referring to their diagnosis occurring "5 or 6 years ago", were not sufficiently precise to be included in the demographic characteristics table.

R3_4. *Were all the transcripts translated into English? Did any of the researchers read all transcripts? This could be clearer.*

Thank you for highlighting this ambiguity. The non-English language interviews were conducted by a native speaker research assistant rather than author 1 to facilitate the flow of conversation and to ensure that language would not be a barrier to comprehension for either party. However, the first author was able to read all the transcripts in their original language and conducted the initial stages of thematic analysis. Sections of the non-English transcripts were translated into English by the first author in collaboration with a native speaker, to facilitate discussion and analysis amongst the other authors. We have clarified this in the text (p. 5).

Reviewer 3

The authors have done an excellent job of responding to the reviewers' comments and revising their manuscript accordingly.

We thank the reviewer for this positive evaluation of the revisions.

I have three additional comments for the authors to consider.

R3_1 *First, while the aim of this paper was to understand how older autistic adults experience and define quality of life, the discussion could benefit from the authors being more explicit about whether the current findings corroborate the findings from existing studies, mostly on younger autistic adults. That is to say, are the five themes identified – Diagnosis is pivotal, Connection with others, Autonomy over space and time, 'Paperwork of life', and Vulnerability – characteristic of autistic quality of life across the lifespan or are there any specific findings /aspects of quality of life that appear to be particularly important for (or unique to) older autistic adults? The authors do make these comparisons at some points (e.g., discrimination may be exacerbated with increasing age, as individuals encounter ageism, p. 13), but more explicit discussion of this issue would be helpful.*

We have ensured that the themes contain references to the existing literature across the lifespan, including with younger autistic adults (pp. 11-12). We have also added a comment in the discussion highlighting how, while certain themes may be echoed across age groups, they may gain or lose relevance based on the changing contexts of adulthood (p.13) . Finally we have emphasised the need for longituinal work to assess whether there are changes in QoL across the lifespan (p.14).

R3_2 *Second, on the theme titled, 'paperwork of life', much has been written on the overly bureaucratic nature of benefits systems and the stigma it can cause disabled people (e.g. see Saffer et al., 2018, Disability & Society). There is also some work on how this (relentless) form-filling can be particularly disadvantageous for disabled people – and therefore autistic people – because too many resources (executive functions) are required to navigate the systems necessary to secure such assistance (see Christensen et al., 2020, Public Administrative Review). The discussion of this theme (see p. 13) would be strengthened by reference to this work.*

Thank you for highlighting this area of research. We have included a new paragraph on this topic, referencing the suggested studies (p. 12).

R3_3 *Finally, for the new paragraph on pp. 13 and 14 beginning "Reviewing our findings through the lens of the neurodiversity paradigm...", it would be helpful to include some references, not least to demonstrate how the current findings build upon existing theoretical and empirical work.*

We have added relevant references to this statement (p. 13).